# Detection of Neutralizing Antibodies in COVID-19 Patients from Steve Biko Academic Hospital Complex: A Pilot Study

Mankgopo Kgatle [1,2,*], Joseph Musonda Chalwe [3], Donald van der Westhuizen [4], Shuting Xu [5], Botle Precious Damane [6], Precious Mathebela [6], Veronica Ueckermann [7], Simnikiwe Mayaphi [8,9], Hosana Gomes Rodrigues [10], Pedro Moura-Alves [11,12], Honest Ndlovu [1,2,13], Yonwaba Mzizi [1,2], Lusanda Zongo [1,14], Henry Hairwadzi [15], Mariza Vorster [1,2], Jan Rijn Zeevaart [1,16] and Mike Sathekge [1,2,13,*]

1   Nuclear Medicine Research Infrastructure (NuMeRI), Steve Biko Academic Hospital, Pretoria 0002, South Africa; ndlovuhonest@gmail.com (H.N.); yonwabavalentine45@gmail.com (Y.M.); lusanda.zongo@yahoo.com (L.Z.); mariza.vorster@gmail.com (M.V.); janrijn.zeevaart@necsa.co.za (J.R.Z.)
2   Department of Nuclear Medicine, University of Pretoria & Steve Biko Academic Hospital, Pretoria 0001, South Africa
3   AXIM Nuclear and Oncology, Laboratory & Scientific Division, 63 Old Pretoria Road, Midrand 1685, South Africa; josephc@axim.co.za
4   Inqaba Biotec, 525 Justice Mohamed Rd, Pretoria 0002, South Africa; donald.vanderwesthuizen@inqababiotec.co.za
5   GenScript Biotech (Netherlands) BV., 2288 EG Rijswijk, The Netherlands; shuting_xu@yahoo.com
6   Department of Surgery, University of Pretoria & Steve Biko Academic Hospital, Hatfield 0028, South Africa; precious.setlai@up.ac.za (B.P.D.); mathebela.precious@up.ac.za (P.M.)
7   Department Internal Medicine, University of Pretoria & Steve Biko Academic Hospital, Pretoria 0001, South Africa; veronica.ueckermann@up.ac.za
8   Department of Medical Virology, University of Pretoria, Tshwane 0002, South Africa; sim.mayaphi@up.ac.za
9   National Health Laboratory Service-Tshwane Academic Division (NHLS-TAD), Tshwane 0002, South Africa
10  Laboratório de Nutrientes e Reparo Tecidual, Faculdade de Ciências Aplicadas, Universidade Estadual de Campinas, Limeira 13484-350, SP, Brazil; hosana.rodrigues@fca.unicamp.br
11  IBMC-Instituto de Biologia Molecular e Celular, Universidade do Porto, 4200-135 Porto, Portugal; pmouraalves@i3s.up.pt
12  i3S-Instituto de Investigação e Inovação em Saúde, Universidade do Porto, 4200-135 Porto, Portugal
13  Department of Nuclear Medicine, Steve Biko Academic Hospital, Pretoria 0001, South Africa
14  Faculty of Health Science, University of Pretoria, Pretoria 0001, South Africa
15  Division of Hepatology and Liver Research, Department of Medicine, University of Cape Town, Groote Schuur Hospital, Cape Town 7925, South Africa; henry.hairwadzi@gmail.com
16  South African Nuclear Energy Corporation, Radiochemistry, Elias Motsoaledi Street, R104 Pelindaba, Northwest 0240, South Africa
*   Correspondence: mankgopo.kgatle@sanumeri.co.za or kgatle.mankgopo@gmail.com (M.K.); mike.sathekge@up.ac.za (M.S.)

**Abstract:** A correlation between neutralization activity after severe acute respiratory syndrome coronavirus 2 (SARS-CoV-2) vaccination and protection against coronavirus disease 2019 (COVID-19) has been demonstrated by several studies. Here, we detect SARS-CoV-2 neutralizing antibody (NAB) production in COVID-19 patients from the Steve Biko Academic Hospital complex (SBAH), South Africa (SA). Samples from COVID-19 patients (mild to severe) were collected. SARS-CoV-2 rapid assays, genotyping (Delta and Omicron variants) and enzyme-linked immunosorbent assays (ELISA) were performed. IBM® Statistical Package for the Social Sciences (SPSS®) version 28 was used for inferential statistical analysis, and the data were presented using the Prism9 software (version 9.4.1). A total of 137 laboratory-confirmed COVID-19 patients, 12 vaccine recipients and 8 unvaccinated participants were evaluated. The production of SARS-CoV-2 NABs was observed in some of the COVID-19 cases, mainly in severe cases, although this should be noted with caution due to the small sample size of this pilot study. NABs were also observed in asymptomatic participants, with the most being found in recipients (*n* = 6) of the BNT162b2 (Pfizer-BioNTech) COVID-19 vaccine. We found a strong presence of NABs in COVID-19 patients, specifically in mild and severe cases. Severe infection was associated with higher NAB production (82%).

**Keywords:** neutralizing antibodies; ELISA; SARS-CoV-2; COVID-19; Steve Biko Academic Hospital; South Africa

## 1. Introduction

Coronavirus disease 2019 (COVID-19) is an infectious disease that occurs as a result of infection with severe acute respiratory syndrome coronavirus 2 (SARS-CoV-2) from the Coronaviridae family in the Nidovirales [1]. The World Health Organization (WHO) declared COVID-19 as a pandemic on 11 March 2020. As of 7 June 2023, more than 767 million cases and 6.9 million deaths have been reported worldwide. COVID-19 develops a range of symptoms that vary over a period of time with the progression of the disease. The most prominent symptoms are fever, cough, fatigue, dyspnea and loss of taste or smell. COVID-19 has been reported to persist and lead to prolonged illness [2]. Infection by the SARS-CoV-2 virus can be classified into three stages: stage I, which is an asymptomatic incubation period, where the virus may or may not be detected; stage II, where there may be a moderate symptomatic period to a peak viral load incidence; stage III, a severe stage with respiratory symptoms and an elevated viral load [3]. An essential aspect of COVID-19 is that some positive cases remain asymptomatic, presenting a challenge for total patient detection and transmission control [4]. Antibodies for SARS-CoV-2 are produced in a period of 7 days after the beginning of the array of symptoms. The first type of antibodies to be produced (IgM) may appear earlier (5 days) and decline rapidly within a few weeks. The second type of antibodies (IgGs) is extremely potent, with a unique ability to link foreign particles to innate immune cells, which are specific to the viral infection. The IgG concentrations persist for a longer period of at least 3 months. Neutralizing antibodies (NABs) are crucial for specific defense against viral invaders [5]. They do not only attach to the virus, but they might also inhibit interactions with the receptor or attach to a viral capsid in such a way that prevents the uncoating of the genome, hence helping in infection control [5]. Proteins located on the surfaces of viruses initiate the activity of NABs [6]. Published evidence suggests that insights into how clinical characteristics affect humoral immunity is crucial in the understanding of COVID-19 [7]. The production of antibodies in response to the receptor-binding domain (RBD) of the SARS-CoV-2 spike protein is vital for the development of immunological mechanisms of protection [8]. A number of factors have been reported to affect the humoral responses to infection by SARS-CoV-2 and vaccination in several studies [9–11]. These factors include advanced age [9], gender [10] and immunosuppression [11]. Despite this, the mechanisms by which disease severity influences humoral immunity—for instance, the production of neutralizing antibodies (NAB)—is still unclear [12].

Current studies have shown that NABs are present in the system approximately 2 weeks from the onset of the infection and last for a minimum of 4 weeks. However, it is still not clear whether all patients produce NABs, whether there is an association between their titers and disease severity or whether their NAB concentrations are sufficient to induce the innate humoral immune response to the infection [13,14]. Chvatal-Medina et al. [11] reported that vaccine efficacy, effectiveness and antibody-dependent enhancement (ADE) against SARS-CoV-2 are areas that still require investigation. In line with this, we aimed to detect SARS-CoV-2 infection in our study participants, which included COVID-19 patients and healthy vaccine recipients at Steve Biko Academic Hospital (SBAH) in South Africa (SA), and their NAB production.

## 2. Materials and Methods

### 2.1. Study Population and Specimen Collection

A convenience sampling method was used, and the 157 participants who gave consent were recruited to participate in this pilot study at the SBAH Complex (SBAH and Tshwane District Hospital) in Pretoria, SA. A total of 137 laboratory-confirmed COVID-19 patients,

12 vaccine recipients and 8 unvaccinated participants were enrolled between December 2021 and May 2022. During this time, the Omicron variant of SARS-CoV-2 was dominant in South Africa. A nasopharyngeal swab in RNA/DNA Shield reagent and blood in a serum tube were collected by a qualified phlebotomist registered with the Health Professions Council of South Africa (HPCSA) using standardized techniques. Good laboratory practice (GLP) procedures were followed to ensure accurate and reliable results. For the main analysis, the participants were categorized into 6 groups, namely Group 1: COVID-19 mild cases; Group 2: COVID-19 moderate cases; Group 3: COVID-19 severe cases; Group 4: healthy (non-COVID-19) and unvaccinated participants; Group 5: healthy (non-COVID-19) and vaccinated with the Ad26.COV2.S (Johnson & Johnson) COVID-19 vaccine; Group 6: healthy (non-COVID-19) and vaccinated with the BNT162b2 (Pfizer-BioNTech) COVID-19 vaccine. The COVID-19 severity classification defined by the Steve Biko Academic Hospital/Tshwane District Hospital was used and is illustrated in Supplementary Table S1.

## 2.2. RNA Extraction

RNA extraction was performed using the ExtractMe viral RNA kit (Blirt, 172 Gdańsk, Poland—Cat. No. EM39) following the manufacturer's instructions. Briefly, 100 μL of the RNA/DNA shield containing a swab sample was used for extraction, and this was transferred into a 1.5 mL Eppendorf tube. Then, 400 μL of the provided vRLys Buffer was added and the mixture was vortexed for 10 s. The sample was incubated for 5 min at room temperature and vortexed several times during incubation. Next, 250 μL isopropanol was added, and the mixture was inverted several times prior to being transferred into an RNA purification column placed in a collection tube. This was followed by centrifugation for 30 s at $12,000 \times g$, and the supernatant was discarded. Then, 700 μL vRW buffer was added into the column as the first wash and the mixture was centrifuged under the same conditions. The supernatant was discarded and the column was washed again with 300 μL vRW buffer. After being centrifuged twice for 180 s at $15,000 \times g$, the purification minicolumn was transferred into a sterile RNase-free 1.5 mL Eppendorf tube. The purified RNA was eluted with 30 μL elution buffer for 60 s at $12,000 \times g$. The sample RNA concentrations were measured using a Qubit 4 Fluorometer (Thermo Fisher Scientific, Wilmington, DE, USA). There was no inhibition of PCR as there was successful amplification of IC in all samples.

## 2.3. SARS-CoV-2 Genotyping

The extracted samples were sent to Inqaba Biotechnical Industries (Pty), Ltd. (525 Justice Mahommed Road, Muckleneuk, Pretoria, 0002) for SARS-CoV-2 genotyping. Briefly, the samples were subjected to the AllPlex SARS-CoV-2 Master Assay (Seegene, Seoul, Republic of Korea), which detects four SARS-CoV-2 target genes and five notable S gene mutations (N501Y, HV69/70 del, P681H, which are specific to Omicron, and E484K, Y144de).

The manufacturer's instructions were followed in the PCR preparation as follows. A master mix was prepared according to the number of samples, plus a positive and negative control. Per sample, 5 μL of SC2M MuDT oligo mix, 5 uL of enzyme mix and 5 uL of RNAse-free water were combined. This was distributed to wells on a white qPCR plate (BioRad HSP9655). Then, 5 uL of extracted sample was added to the master mix. The sample was then placed on a Bio-Rad CFX96 with the following cycling conditions: 50 degrees for 20 min; 95 degrees for 15 min; 45 cycles of 95 degrees for 10 s, 60 degrees for 15 s with plate read, 72 degrees for 10 s with plate read. The same samples were then reflex-tested on the AllPlex SARS-CoV-2 Variants II panel (Seegene, Seoul, Republic of Korea) to identify four notable SARS-CoV-2 S gene mutations (L452R, W152C, K417T and K417N) and distinguish between the B. 1.617. 2 (Delta) variant and B.1.1.529 (Omicron) variant. The manufacturer's instructions were followed as described above. The samples were then placed on a Bio-Rad CFX96 with the following cycling conditions: 50 degrees for 20 min; 95 degrees for 15 min; 3 cycles of 95 degrees for 10 s, 60 degrees for 40 s, 72 degrees for 10 s; 42 cycles of 95 degrees for 10 s, 60 degrees for 15 s with plate read and 72 degrees for 10 s with plate read.

The absence of the S variant signal in the Master Assay and the presence of L452R indicated the Delta variant. The presence of the S variant signal in the Master Assay indicated the Omicron variant. The Delta and Omicron strains were prevalent and were the variants of concern (VOCs) during the time of this study.

### 2.4. COVID-19 IgG/IgM Rapid Test

The detection of SARS-CoV-2 IgG/IgM antibodies was carried out using the single-prick COVID-19 IgG/IgM Rapid Test Cassettes from Orient Gene Biotech, according to the manufacturer's instructions (Orient Gene Biotech, Huzhou, Zhejiang, China). A test cassette was aseptically removed from the sealed foil pouch and used immediately. Briefly, a test cassette was placed on a clean and level surface. The participant's finger was pricked with a needle and a drop of blood was drawn with a plastic dropper and transferred to the specimen well of a test cassette. Two drops of sample buffer, provided with the kit, were added immediately into the buffer well of the test cassette. The result was read within 10 min. In the vast majority of the tests, positive results became visible within 2 min.

### 2.5. SARS-CoV-2 Neutralizing Antibody Detection ELISA Assay

All tests were performed in duplicate using the cPass™ SARS-CoV-2 Neutralization Antibody Detection Kit from Nanjing GenScript Biotech Co., Ltd. (Nanjing, Jiangsu Province, China) [15], according to the manufacturer's instructions. Diluted positive control, diluted negative control and diluted samples were mixed with diluted HRP-RBD at a volume ratio of 1:1 in tubes and then incubated at 37 °C for 30 min. Then, 100 µL of the positive control mixture, the negative control mixture and the sample mixture was added to the corresponding wells and incubated at 37 °C for 15 min. The plate was washed with 260 µL of 1× wash solution per well four times using the W206 Microplate Washer (Chengdu Empsun Medical Technology Co., Ltd., Sichuan, China). Then, 100 µL of TMB solution was added to each well and the plate was incubated in the dark at 20 to 25 °C for 15 min. Next, 50 µL of stop solution was added to each well to stop the reaction. The plate was immediately read using the M201 ELISA microplate reader (Chengdu Empsun Medical Technology Co., Ltd., Sichuan, China). The average optical density (OD) of the negative control was used to calculate the inhibition percentage. The results of each sample were calculated using the formula below:

$$\text{Inhibition} = \{1 - (\text{OD value of sample}/\text{OD value of negative control})\} \times 100\%$$

Sample results were interpreted as follows:

- [≥30%] = a positive result, indicating the presence of SARS-CoV-2 neutralizing antibodies;
- [<30%] = a negative result, indicating the absence of SARS-CoV-2 neutralizing antibodies, or that the detected level was below the limit of detection.

### 2.6. Statistical Analysis

All the raw data were captured in Microsoft Office Excel (Microsoft Corp, Redmon, WA, USA) and exported to IBM® Statistical Package for the Social Sciences (SPSS®) version 28 for inferential statistical analysis (IBM Corp, Armonk, NY, USA). All *p*-values were two-tailed, and those <0.05 were considered statistically significant. Only significant correlations are reported as results. Means and SDs were used for normally distributed data and medians (interquartile range (IQR)) for data that were not normally distributed. Spearman's correlation analysis was performed between the different parameters and NAB production. For data presentation, the Prism9 software (version 9.4.1) was used.

## 3. Results

The clinical characteristics of the COVID-19 patients are outlined in Table 1. The ratio of male to female participants was balanced, with 69 out of the total 137 patients being female (50.4%) and 68 being male (49.6%).

**Table 1.** General characteristics of the patients.

| | Mild | Moderate | Severe | r-Value | p-Value |
|---|---|---|---|---|---|
| | No.(%) | No.(%) | No.(%) | | |
| Age (years) | | | | | |
| Total | 48 | 42 | 47 | −0.01694 | 0.84424 |
| <40 | 10 (20.8) | 7(16.7) | 12(25.5) | | |
| 40–69 | 37(77) | 26(61.9) | 30(63.8) | | |
| ≥70 | 1(2) | 9(2.1) | 5(10.6) | | |
| Gender | | | | | |
| Male | 28(58.3) | 19(45.2) | 21(44.7) | −0.11057 | 0.19836 |
| Female | 20(41.7) | 23(54.8) | 26(55.3) | | |
| Smoker | | | | | |
| Yes | 11(22.9) | 7(16.7) | 44(93.6) | | |
| No | 34(70.8) | 32(76.2) | 3(6.4) | 0.17099 | 0.04574 |
| Ex-smoker | 3(6.3) | 3(7.1) | 0(0) | | |
| Variants | | | | | |
| Beta | | | | | |
| Delta | 27(56.3) | 22(52.4) | 34(72.3) | −0.00567 | 0.94761 |
| Omicron | 21(43.8) | 20(47.6) | 13(27.7) | | |
| Vaccination | | | | | |
| Yes | 6(12.5) | 9(21.4) | 4(8.5) | 0.08305 | 0.33465 |
| No | 42(87.5) | 33(78.6) | 43(91.5) | | |
| Co-morbidities | | | | | |
| HIV | | | | | |
| Yes | 9(18.8) | 6(14.3) | 6(12.8) | −0.05431 | 0.52849 |
| No | 39(81.3) | 36(85.7) | 41(87.2) | | |
| Hypertension | | | | | |
| Yes | 19(39.6) | 18(42.9) | 26(55.3) | 0.11351 | 0.18659 |
| No | 29(60.4) | 24(57.1) | 21(44.7) | | |
| Diabetes | | | | | |
| Yes | 7(14.6) | 9(21.4) | 18(38.3) | −0.03568 | 0.67892 |
| No | 41(85.4) | 33(78.6) | 29(61.7) | | |

A total of 137 laboratory-confirmed COVID-19 patients with various clinical statuses, 12 vaccine recipients and 8 unvaccinated participants were evaluated (Figure 1). The median age for all the participants was 58 years, with an interquartile range (IQR) of 23. The minimum age was 24 and the maximum age was 80.

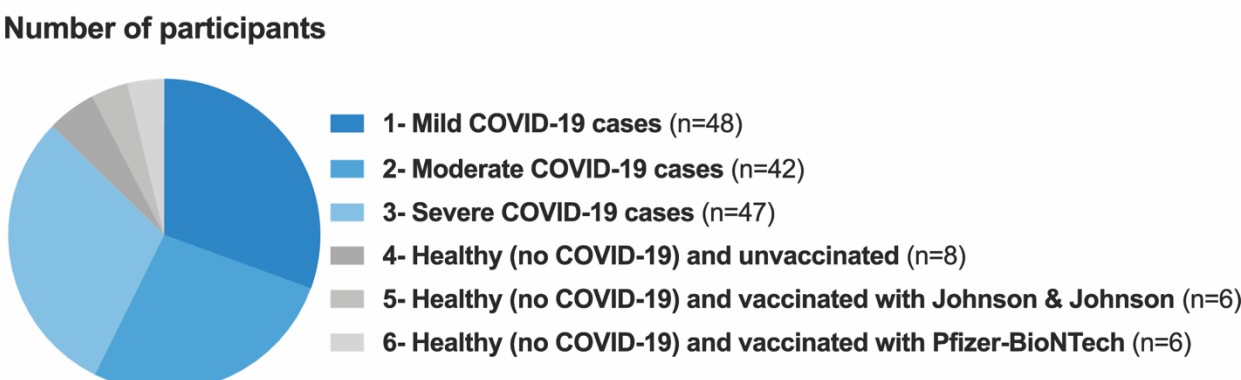

**Figure 1.** Number of participants.

Of the 137 COVID-19 patients recruited for this study, 83 (60.6%) were infected with the Delta variant, whereas 54 (39.4%) were infected with the Omicron variant (Table 1). Out of the 137 samples, 19 (13.9%) were vaccinated individuals (Table 1), and the type of vaccine and distribution is shown in Figure 2.

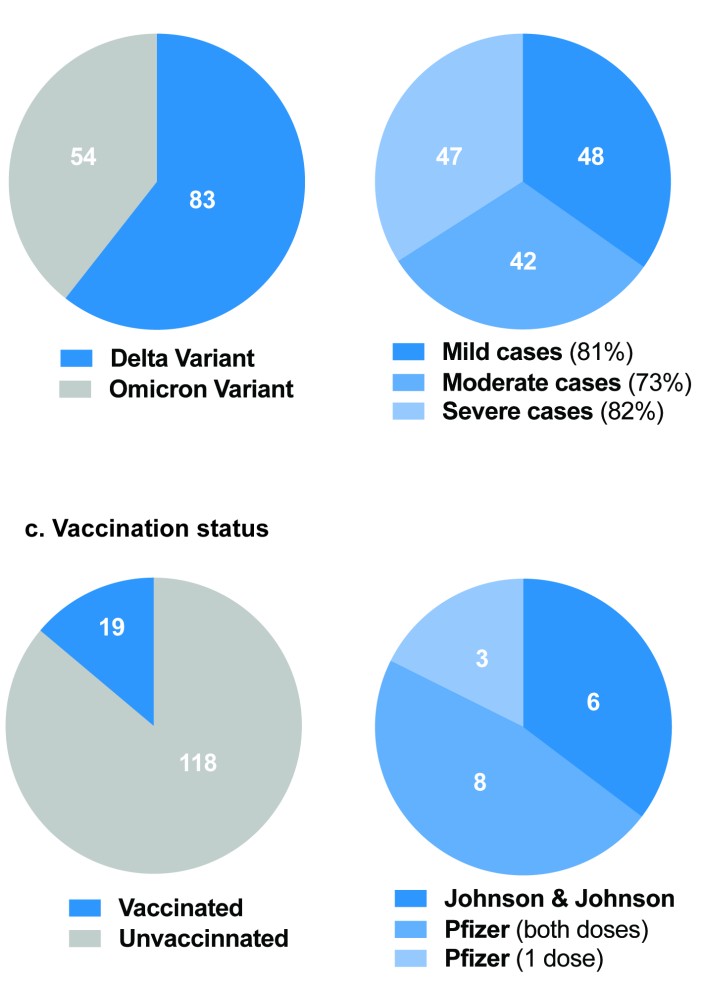

**Figure 2.** COVID-19 variants and vaccination status of patients.

The production of SARS-CoV-2 NABs was observed in 81% of mild cases, 73% of moderate cases and 82% of severe cases (Figure 3).

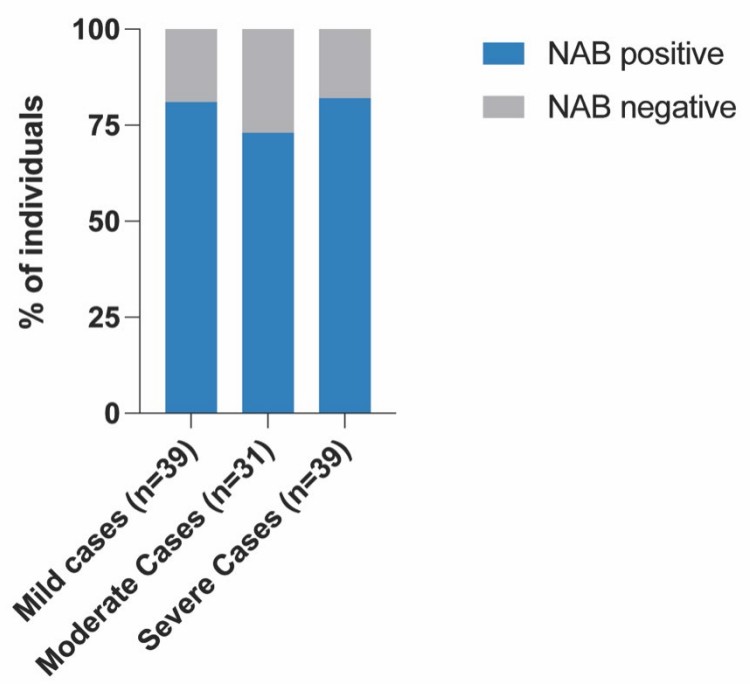

**Figure 3.** NAB production in mild to severe COVID-19 cases.

NABs were also observed in some healthy participants: 25% in unvaccinated participants, 33% in Johnson & Johnson vaccine recipients and all (100%) Pfizer-BioNTech vaccine recipients (Figure 4).

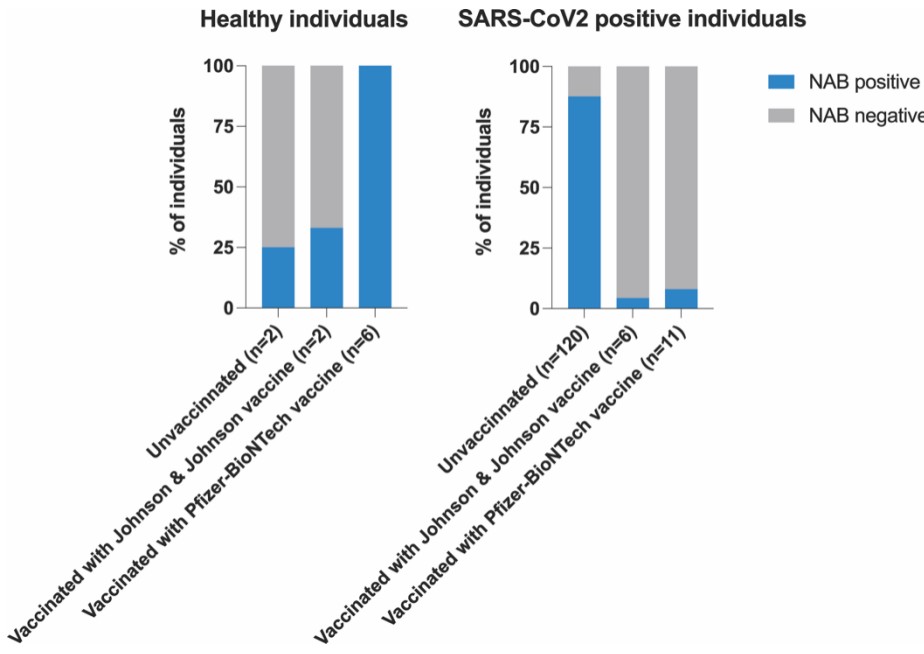

**Figure 4.** NAB production in SARS-CoV-2-positive and healthy individuals.

*Correlation between NABs, COVID-19 Variants and Healthy Participants*

There was no significant positive correlation observed between the different parameters in Table 2 and NAB production ($r/p$) (two-tailed).

**Table 2.** Spearman's correlation between the different parameters and NAB production (*r/p*).

| NAB Production | *r* | *p* |
|---|---|---|
| Age | −0.01694 | 0.84424 |
| Gender | 0.05021 | 0.56013 |
| VOCs | −0.00567 | 0.94761 |
| Mild | −0.01061 | 0.94292 |
| Moderate | 0.1574 | 0.31948 |
| Severe | −0.31394 | 0.03164 |

## 4. Discussion

Infection with SARS-CoV-2 initiates an adaptive immune response, which includes the production of antibodies that can be detected in the blood of infected patients. The secreted antibodies provide protection against future infections from viruses as they remain in the circulatory system for months to years after infection. A subpopulation of circulating antibodies (NABs) can block the cellular infiltration and replication of the virus [6].

In our study, NABs were detected in individuals of all six groups, as shown in Figure 1. Of note, similar to the findings of Padoan et al. [16] and not aligned with other reports on the effect of age (*p* = 0.8442) and gender on the production of NABs, we did not see any significant positive relationship between NAB production and gender (*p* = 0.56013), nor did we see a significant relationship between NAB production and the two variants of concern (VOCs) (*p* = 0.94761) investigated in this study. The group with severe cases of COVID-19 showed the highest presence of NABs (82%). Similar findings were reported by Garcia-Beltran et al. [17] in Boston, United States of America (USA), in 113 COVID-19 cases. This is in line with previous studies, showing a clear relationship between COVID-19 severity and the development of humoral immunity [18,19]. Although this humoral immune response is mediated by NABs, the cause of this association with severe COVID-19 is still being investigated. Similarly, others [17] have proposed that one reason for results such as those observed in this study may be the hyperinflammation and uninhibited viral replication observed in severe COVID-19 cases, which can lead to antibodies' overproduction. The group with mild cases of COVID-19 in our study also had a high prevalence of NABs (81%). This is also in line with other studies showing that individuals develop NABs to SARS-CoV-2 within a few days of infection, although these NABs may decline over time [6,20]. The group with moderate cases of COVID-19 in our study only had an NAB prevalence of 73%. Whilst some researchers have agreed on the fact that humoral immunity is sustained, others, such as Pang et al. [6], have reported that this immunity to SARS-CoV-2 may be brief in patients with moderate cases, which could be the case in our study population.

Vaccines are designed to train the immune system and boost its response to combat particular viruses, such as SARS-CoV-2. COVID-19 vaccines and boosters confer immune protection by stimulating T-cell responses and NAB titers, preventing severe clinical outcomes, hospitalization and related fatalities [21]. Several studies have reported on the association between NABs after SARS-CoV-2 vaccination and protection against COVID-19 [16,22–26]. Importantly, a singular characteristic of our study population is the fact that the majority of our COVID-19 patients were not vaccinated (87.6%). Nevertheless, and although a small population of our participants were vaccinated, we still observed some breakthrough infection cases (*n* = 5). A breakthrough infection occurs when a vaccinated person becomes infected with COVID-19 [26]. Our findings show that most of the vaccinated COVID-19 patients either had mild or moderate symptoms, which is consistent with the reports by Reynolds et al. [26] and McIntyre et al. [27]. Interestingly, two out of the six patients who had received the Johnson & Johnson vaccine developed severe cases of COVID-19. Another patient who also received the Johnson & Johnson vaccine developed a mild case of COVID-19. Moreover, one out of the three patients who had received a single

dose of the Pfizer-BioNTech vaccine developed a moderate case of COVID-19. One of the patients who had received both doses of the Pfizer-BioNTech vaccine also developed a moderate case of COVID-19. In addition to this, only 64.7% of the vaccinated patients produced NABs. A local study by Madhi et al. [28] also indicated that vaccinated participants were more likely to show seropositivity for SARS-CoV-2 than unvaccinated participants. Overall, with subtle differences between the different vaccination regimens, our results confirm the efficacy of vaccination in preventing severe disease outcomes in most patients. In brief, 87.5% of the patient group that had been vaccinated with the Pfizer-BioNTech vaccine had NABs against SARS-CoV-2, whereas only 33.3% of the patients that were vaccinated with the Johnson & Johnson vaccine had NABs. Hence, in our study, the Pfizer-BioNTech vaccine showed the stronger production of NABs in our COVID-19 patients.

The group with healthy participants that had been vaccinated with the Johnson & Johnson vaccine in our study showed a lower prevalence (33%) of NABs against SARS-CoV-2. This is parallel to a report by Tada et al. [23], who also found that the Johnson & Johnson vaccine NABs showed a noticeable decline in neutralizing titers against SARS-CoV-2, which may lead to the possibility of decreased protection against VOCs [23]. Lastly, but importantly, our results show that healthy participants that had been vaccinated with the Pfizer-BioNTech vaccine all had NABs against SARS-CoV-2 (100%). Our findings are similar to those of Padoan et al. [16], who reported that the Pfizer-BioNTech vaccine stimulates the robust production of NABs, particularly 28 days after the first inoculum.

The correlation between the production of NABs and the severity of a disease in an individual is still unclear. Recent evidence has, however, suggested some possible clarifications of this link. For example, the presence of hyperinflammation, which is described as acute inflammation with a cytokine storm, can increase disease severity despite the viral load [29–32]. Furthermore, an elevated viral load has also been shown to escalate the severity of a disease, which eventually increases the production of antibodies [33,34]. Although NABs have been measured in vaccine recipients for most of the approved vaccines for COVID-19, different methods have been employed in these studies, which presents a challenge when directly comparing them. The majority of these reports have, however, indicated a favorable humoral response for the antibodies [35,36].

Arankalle and co-authors [36] have reported that vaccines in conjunction with natural immunity disseminate NABs, which provide an immune response and reduce breakthrough infections. Mounting evidence [37,38] also demonstrates that infection by SARS-CoV-2 is possible in both vaccinated and unvaccinated individuals. The production of NABs offers protection from reinfections and minimizes the likelihood of progression to severe disease. However, with the discovery of new VOCs, such as the sub-lineage of Omicron [39], vaccinations continue to be the best measure for the control of COVID-19.

Interestingly, in our study, some of the unvaccinated healthy participants had NABs (25%). This may have simply been a false positive that may be decreased with an increase in the sample size. Nonetheless, Alejo et al. [18] and Oh et al. [19] reported similar findings in their populations and attributed this observation to possible natural immunity. Although this immunity is not fully understood, this might also have been the case in our small population of participants.

## 5. Conclusions

In conclusion, we found a strong presence of NABs in several of the COVID-19 patients, specifically in mild and severe cases. Severe infection was associated with higher NAB production. With respect to vaccination and protection, the vaccinated participants had a significantly higher NAB percentage, specifically with the Pfizer-BioNTech vaccine compared to the Johnson & Johnson vaccine and our groups of COVID-19 patients. Our findings indicate the possibility of stronger humoral immunity caused by the Pfizer-BioNTech vaccine compared to the Johnson & Johnson vaccine.

## 6. Recommendations

We recommend further studies with more participants to better evaluate the statistically significant differences between the diverse populations. Furthermore, we also recommend measuring the antibody titers longitudinally, collecting samples at different times, including symptom onset after infection, to identify SARS-CoV-2 neutralization.

## 7. Limitations

This study had some limitations. Firstly, this study utilized a qualitative methodology instead of a quantitative or semi-quantitative methodology to measure and compare serum-neutralizing antibodies. Additionally, it was carried out on a relatively small number of participants attending the SBAH Complex in Pretoria, SA. The convenient sampling method employed is the reason for the small population size, affecting the generalizability of the findings. Furthermore, the healthy participants were only tested with rapid antibody tests and were not confirmed with quantitative polymerase chain reaction (qPCR) detection. For this reason, the findings in this report cannot be generalized to the broader population but only applied to our setting.

**Supplementary Materials:** The following supporting information can be downloaded at: https://www.mdpi.com/article/10.3390/covid3070072/s1. Table S1: COVID-19 Severity Classification: Steve Biko Academic Hospital Complex.

**Author Contributions:** M.K., M.S., P.M., H.G.R., M.V., J.R.Z. and H.H. conceptualized and designed the experiments. M.K., Y.M., L.Z., V.U. and S.M. contributed to the collection and processing of the samples and designed some experiments. M.K., J.M.C., B.P.D., P.M., H.N., Y.M., S.X. and L.Z. conducted the experiments. J.M.C., D.v.d.W. and P.M.-A. analyzed the data. All the authors contributed to the writing and review of the manuscript. All authors have read and agreed to the published version of the manuscript.

**Funding:** This research was self-funded by NUMERI and NRF (Grant No.: 138154).

**Institutional Review Board Statement:** This study was approved by the University of Pretoria Research Ethics Committee (Ref No.: 28/2021), ensuring participant autonomy, beneficence and non-maleficence. Participation in this study was voluntary and participants gave consent. The study was conducted in accordance with the Declaration of Helsinki and approved by the Ethics Committee for studies involving humans.

**Informed Consent Statement:** Informed consent was obtained from all subjects involved in the study. Written informed consent has been obtained from the subjects to publish this paper.

**Data Availability Statement:** The data from this study are available from the authors upon request.

**Acknowledgments:** P.M thanks H2020-WIDESPREAD-2018-951921-ImmunoHUB for the financial support.

**Conflicts of Interest:** The authors declare no conflict of interest.

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
