# Peer review of "Detection of Neutralizing Antibodies in COVID-19 Patients from Steve Biko Academic Hospital Complex: A Pilot Study"

_covid, doi:10.3390/covid3070072_

Round 1

Reviewer 1 Report

The production of NAB after SARS-CoV-2 infection is a very important measurement method that can be used as a very important indicator of the group's defensive immunity as well as the treatment of patients.
However, the number of 12 vacine recipients and 8 un-34 vacated partners seems to be insufficient to make statistical analysis difficult.

Author Response

Comment: The production of NAB after SARS-CoV-2 infection is a very important measurement method that can be used as a very important indicator of the group's defensive immunity as well as the treatment of patients. However, the number of 12 vacine recipients and 8 un-34 vacated partners seems to be insufficient to make statistical analysis difficult.

Response

We appreciate this important comment. We have addressed this by adding the statement below in abstract:

“The production of SARS-CoV-2 NABs was observed in some of the COVID-19 cases, mainly in severe cases, although this should be noted with caution due to the small sample size of this pilot study.”

Additionally, we have improved the manuscript by adding more information for clarity in the introduction, the methods and results.

Reviewer 2 Report

In this study, the authors have characterized the neutralizing antibody response in SARS-CoV-2-infected patients. The manuscript is unclear in explaining things and inconsistencies in the text. The following comments need to be addressed.

1. In line 35, the authors mentioned that neutralizing antibodies were observed in some of the COVID-19 cases, mainly in severe cases, although it was not statistically significant. However, in line 38, it is mentioned that a strong presence of neutralizing antibodies in COVID-19 patients, specifically in mild and severe cases. These statements contradict each other.

2. There were several mismatches between Table 1 and the figures.

For example

a. In table 1 – vaccination 6+10+5= Total 21. In Figure 2C, it is 17.  

b. In table 1 – delta 28+20+34= Total 82. In Figure 2A, it is 84.  

c. In table 1 – omicron 20+22+13= Total 55. In Figure 2A, it is 53.  

d. Same for disease status as well.

I suggest the authors should review the manuscript carefully. 

Author Response

Comment 1

In this study, the authors have characterized the neutralizing antibody response in SARS-CoV-2-infected patients. The manuscript is unclear in explaining things and inconsistencies in the text. The following comments need to be addressed.

In line 35, the authors mentioned that neutralizing antibodies were observed in some of the COVID-19 cases, mainly in severe cases, although it was not statistically significant. However, in line 38, it is mentioned that a strong presence of neutralizing antibodies in COVID-19 patients, specifically in mild and severe cases. These statements contradict each other.

Response: Thanks for this important comment. To address this comment, we have modified this sentence in the abstract to read this way: “The production of SARS-CoV-2 NABs was observed in some of the COVID-19 cases, mainly in severe cases, although this should be noted with caution due to the small sample size of this pilot study.”

Comment 2: There were several mismatches between Table 1 and the figures.

For example, In table 1 – vaccination 6+9+4= Total 19. Non-vaccination 42+33+43 = 118. In Figure 2C, it is 17.  In table 1 – delta 27+22+34= Total 83. In Figure 2A, it is 84. In table 1 – omicron 21+20+13= Total 54. In Figure 2A, it is 53.  Same for disease status as well. I suggest the authors should review the manuscript carefully. 

Response: we sincerely apologize for this error, and fixed the entire Table 1 and the Figures. They now correspond.

Round 2

Reviewer 2 Report

The authors have addressed all my concerns.